# LEARNING TO REPRESENT EDITS

**Pengcheng Yin,**[*] **Graham Neubig**
Language Technology Institute
Carnegie Mellon University
Pittsburgh, PA 15213, USA
{pcyin,gneubig}@cs.cmu.edu

**Miltiadis Allamanis, Marc Brockschmidt, Alexander L. Gaunt**
Microsoft Research
Cambridge, CB1 2FB, United Kingdom
{miallama,mabrocks,algaunt}@microsoft.com

## ABSTRACT

We introduce the problem of learning distributed representations of edits. By combining a "neural editor" with an "edit encoder", our models learn to represent the salient information of an edit and can be used to apply edits to new inputs. We experiment on natural language and source code edit data. Our evaluation yields promising results that suggest that our neural network models learn to capture the structure and semantics of edits. We hope that this interesting task and data source will inspire other researchers to work further on this problem.

## 1 INTRODUCTION

One great advantage of electronic storage of documents is the ease with which we can edit them, and edits are performed in a wide variety of contents. For example, right before a conference deadline, papers worldwide are finalized and polished, often involving common fixes for grammar, clarity and style. Would it be possible to automatically extract rules from these common edits? Similarly, program source code is constantly changed to implement new features, follow best practices and fix bugs. With the widespread deployment of (implicit) version control systems, these edits are quickly archived, creating a major data stream that we can learn from.

In this work, we study the problem of learning distributed representations of edits. We only look at small edits with simple semantics that are more likely to appear often and do not consider larger edits; *i.e.*, we consider "add definite articles" rather than "rewrite act 2, scene 3." Concretely, we focus on two questions: i) Can we group semantically equivalent edits together, so that we can automatically recognize common edit patterns? ii) Can we automatically transfer edits from one context to another? A solution to the first question would yield a practical tool for copy editors and programmers alike, automatically identifying the most common changes. By leveraging tools from program synthesis, such groups of edits could be turned into interpretable rules and scripts (Rolim et al., 2017). When there is no simple hard rule explaining how to apply an edit, an answer to the second question would be of great use, *e.g.*, to automatically rewrite natural language following some stylistic rule.

We propose to handle edit data in an autoencoder-style framework, in which an "edit encoder" $f_\Delta$ is trained to compute a representation of an edit $x_- \to x_+$, and a "neural editor" $\alpha$ is trained to construct $x_+$ from the edit representation and $x_-$. This framework ensures that the edit representation is semantically meaningful, and a sufficiently strong neural editor allows this representation to not be specific to the changed element. We experiment with various neural architectures that can learn to represent and apply edits and hope to direct the attention of the research community to this new and interesting data source, leading to better datasets and stronger models.

Briefly, the contributions of our paper are: (a) in Sect. 2, we present a new and important machine learning task on learning representations of edits (b) we present a family of

---

[*]Work done as an intern in Microsoft Research, Cambridge, UK.

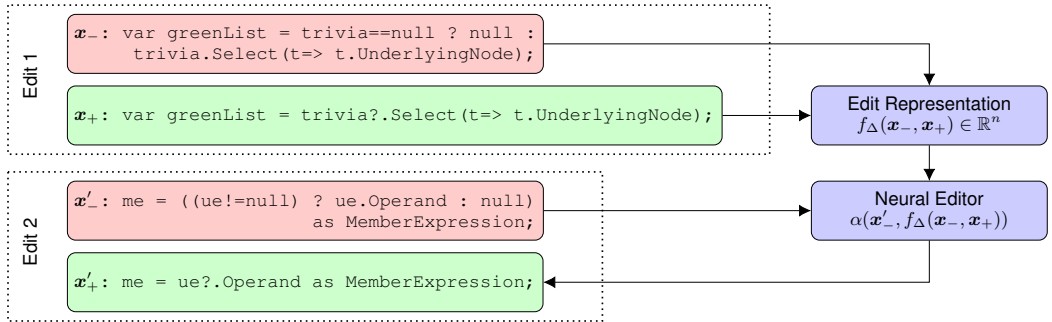

Figure 1: Given an edit (Edit 1) of $\boldsymbol{x}_-$ to $\boldsymbol{x}_+$, $f_\Delta$ computes an edit representation vector. Using that representation vector the neural editor $\alpha$ applies the same edit to a new $\boldsymbol{x}'_-$. The code snippets shown here are real code change examples from the `roslyn` open-source compiler project.

models that capture the structure of edits and compute efficient representations in Sect. 3 (c) we create a new source code edit dataset, and release the data extraction code at `https://github.com/Microsoft/msrc-dpu-learning-to-represent-edits` and the data at `http://www.cs.cmu.edu/~pengchey/githubedits.zip`. (d) we perform a set of experiments on the learned edit representations in Sect. 4 for natural language text and source code and present promising empirical evidence that our models succeed in capturing the semantics of edits.

## 2 TASK

In this work, we are interested in learning to represent and apply edits on discrete sequential or structured data, such as text or source code parse trees[1]. Figure 1 gives a graphical overview of the task, described precisely below.

**Edit Representation** Given a dataset of edits $\{\boldsymbol{x}_-^{(i)} \to \boldsymbol{x}_+^{(i)}\}_{i=1}^N$, where $\boldsymbol{x}_-^{(i)}$ is the original version of some object and $\boldsymbol{x}_+^{(i)}$ its edited form (see upper half of Figure 1 for an example), our goal is to learn a representation function $f_\Delta$ that maps an edit operation $\boldsymbol{x}_- \to \boldsymbol{x}_+$ to a real-valued *edit representation* $f_\Delta(\boldsymbol{x}_-, \boldsymbol{x}_+) \in \mathbb{R}^n$. A desired quality of $f_\Delta$ is for the computed edit representations to have the property that semantically similar edits have nearby representations in $\mathbb{R}^n$. Having distributed representations also allows other interesting downstream tasks, *e.g.*, unsupervised clustering and visualization of similar edits from large-scale data (*e.g.* the GitHub commit stream), which would be useful for developing human-assistance toolkits for discovering and extracting emerging edit patterns (*e.g.* new bug fixes or emerging "best practices" of coding).

**Neural Editor** Given an edit representation function $f_\Delta$, we want to learn to apply edits in a new context. This can be achieved by learning a *neural editor* $\alpha$ that accepts an edit representation $f_\Delta(\boldsymbol{x}_-, \boldsymbol{x}_+)$ and a new input $\boldsymbol{x}'_-$ and generates $\boldsymbol{x}'_+$.[2] This is illustrated in the lower half of Figure 1.

## 3 MODEL

We cast the edit representation problem as an autoencoding task, where we aim to minimize the reconstruction error of $\alpha$ for the edited version $\boldsymbol{x}_+$ given the edit representation $f_\Delta(\boldsymbol{x}_-, \boldsymbol{x}_+)$ and the original version $\boldsymbol{x}_-$. By limiting the capacity of $f_\Delta$'s output and allowing the model to freely use information about $\boldsymbol{x}_-$, we are introducing a "bottleneck" that forces the overall framework to not simply treat $f_\Delta(\boldsymbol{x}_-, \boldsymbol{x}_+)$ as an encoder of $\boldsymbol{x}_+$. The main difference from traditional autoencoders is that in our setup, an optimal solution requires to re-use as much information as possible from $\boldsymbol{x}_-$

---

[1]Existing editing systems, *e.g.* the grammar checker in text editors and code refactoring module in IDEs, are powered by domain-specific, manually crafted rules, while we aim for a data-driven, domain-agnostic approach.

[2]We leave the problem of identifying which edit representation $f_\Delta(\boldsymbol{x}_-, \boldsymbol{x}_+)$ to apply to $\boldsymbol{x}'_-$ as interesting future work.

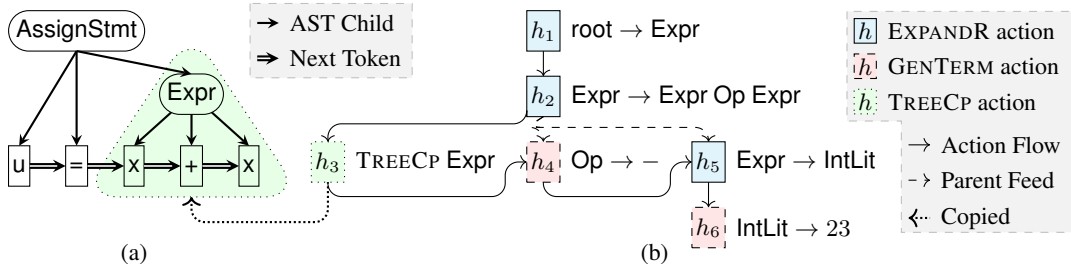

Figure 2: (a) Graph representation of statement `u = x + x`. Rectangular (resp. rounded) nodes denote tokens (resp. non-terminals). (b) Sequence of tree decoding steps yielding `x + x - 23`, where `x + x` is copied (using the TREECP action) from the context graph in (a).

to make the most of the capacity of $f_\Delta$. Formally, given a probabilistic editor function $P_\alpha$ such as a neural network and a dataset $\{x_-^{(i)} \to x_+^{(i)}\}_{i=1}^N$, we seek to minimize the negative likelihood loss

$$\mathcal{L} = -\frac{1}{N} \sum_i \log P_\alpha(x_+ \mid x_-, f_\Delta(x_-, x_+)).$$

Note that this loss function can be interpreted in two ways: (1) as a conditional autoencoder that encodes the salient information of an edit, given $x_-$ and (2) as an encoder-decoder model that encodes $x_-$ and decodes $x_+$ conditioned on the edit representation $f_\Delta(x_-, x_+)$. In the rest of this section, we discuss our methods to model $P_\alpha$ and $f_\Delta$ as neural networks.

## 3.1 NEURAL EDITOR

As discussed above, $\alpha$ should use as much information as possible from $x_-$, and hence, an encoder-decoder architecture with the ability to copy from the input is most appropriate. As we are primarily interested in edits on text and source code in this work, we explored two architectures: a sequence-to-sequence model for text, and a graph-to-tree model for source code, whose known semantics we can leverage both on the encoder as well as on the decoder side. Other classes of edits, for example, image manipulation, would most likely be better served by convolutional neural models.

**Sequence-to-Sequence Neural Editor** First, we consider a standard sequence-to-sequence model with attention (over the tokens of $x_-$). The architecture of our sequence-to-sequence model is similar to that of Luong et al. (2015), with the difference that we use a bidirectional LSTM in the encoder and a token-level copying mechanism (Vinyals et al., 2015) that directly copies tokens into the decoded sequence. Whereas in standard sequence-to-sequence models the decoder is initialized with the representation computed by the encoder, we initialize it with the concatenation of encoder output and the edit representation. We also feed the edit representation as input to the decoder LSTM at each decoding time step. This allows the LSTM decoder to take the edit representation into consideration while generating the output sequence.

**Graph-to-Tree Neural Editor** Our second model aims to take advantage of the additional structure of $x_-$ and $x_+$. To achieve this, we combine a graph-based encoder with a tree-based decoder. We use $T(x)$ to denote a tree representation of an element, *e.g.*, the abstract syntax tree (AST) of a fragment of source code. We extend $T(x)$ into a graph form $G(x)$ by encoding additional relationships (*e.g.*, the "next token" relationship between terminal nodes, *etc.*) (see Figure 2(a)). To encode the elements of $G(x_-)$ into vector representations, we use a gated graph neural network (GGNN) (Li et al., 2015). Similarly to recurrent neural networks for sequences (such as biRNNs), GGNNs compute a representation for each node in the graph, which can be used in the attention mechanisms of a decoder. Additionally, we use them to obtain a representation of the full input $x_-$, by computing their weighted average following the strategy of Gilmer et al. (2017) (*i.e.*, computing a score for each node, normalizing scores with a softmax, and using the resulting values as weights).

Our tree decoder follows the semantic parsing model of Yin & Neubig (2018), which sequentially generate a tree $T(x_+)$ as a series of expansion actions $a_1 \ldots a_N$. The probability of taking an action is modeled as $p(a_t \mid a_{<t}, s)$, where $s$ is the input (a sequence of words in the original semantic

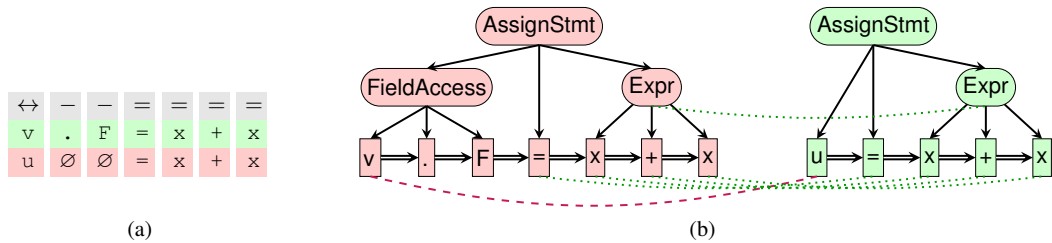

Figure 3: Sequence (a) and graph (b) representation of edit of `v.F = x + x` to `u = x + x`.

parsing setting) and $a_{<t}$ is the partial tree that has been generated so far. The model of Yin & Neubig (2018) mainly uses two types of actions: EXPANDR expands the current non-terminal using a grammar rule, and GENTERM generates a terminal token from a vocabulary or copies a token from $s^3$. The dependence on the partial tree $a_{<t}$ is modeled by an LSTM cell which is used to maintain state throughout the generation procedure. Additionally, the LSTM receives the decoder state used to pick the action at the parent node as an additional input ("parent-feeding"). This process illustrated in Figure 2(b).

We extend this model to our setting by replacing the input sequence $s$ by $\boldsymbol{x}_-$; concretely, we condition the decoder on the graph-level representation computed for $G(\boldsymbol{x}_-)$. Additionally, we use the change representation $f_\Delta(\cdot)$ as an additional input to the LSTM initial state and at every decoding step. Based on the observation that edits to source code often manipulate the syntax tree by moving expressions around (*e.g.* by nesting statements in a conditional, or renaming a function while keeping its arguments), we extend the decoding model of Yin & Neubig (2018) by adding a facility to copy entire subtrees from the input. For this, we add a decoder action TREECP. This action is similar to standard copying mechanism known from pointer networks (Vinyals et al., 2015), but instead of copying only a single token, it copies the whole subtree pointed to.

However, adding the TREECP action means that there are many correct generation sequences for a target tree. This problem appears in token-copying as well, but can be easily circumvented by marginalizing over all correct choices at each generation step (by normalizing the probability distribution over allowed actions to sum up those that have the same effect). In the subtree-copying setting, the lengths of action sequences representing different choices may differ. In our implementation we handle this problem during training by simpling picking the generation sequence that greedily selecting TREECP actions.

## 3.2 EDIT REPRESENTATION

To compute a useful edit representation, a model needs to focus on the differences between $\boldsymbol{x}_-$ and $\boldsymbol{x}_+$. A risk in our framework is that $f_\Delta$ degenerates into an encoder for $\boldsymbol{x}_+$, turning $\alpha$ into a decoder. To avoid this, we need to follow the standard autoencoder trick, *i.e.* it is important to limit the capacity of the result of $f_\Delta$ by generating the edit representation in a low-dimensional space $\mathbb{R}^N$. This acts as a bottleneck and encodes only the information that is needed to reconstruct $\boldsymbol{x}_+$ from $\boldsymbol{x}_-$. We again experimented with both sequence-based and graph-based representations of edits.

**Sequence Encoding of Edits**  Given $\boldsymbol{x}_-$ (resp. $\boldsymbol{x}_+$) as sequence of tokens $t_-^{(0)}, \ldots t_-^{(T_-)}$ (resp. $t_+^{(0)}, \ldots t_+^{(T_+)}$), we can use a standard (deterministic) diffing algorithm to compute an alignment of tokens in the two sequences. We then use extra symbols $\varnothing$ for padding, $+$ for additions, $-$ for deletions, $\leftrightarrow$ for replacements, and $=$ for unchanged tokens to generate a single sequence representing both $\boldsymbol{x}_-$ and $\boldsymbol{x}_+$. This is illustrated in Figure 3(a). By embedding the three entries in each element of the sequence separately and concatenating their representation, they can be fed into a standard sequence encoder whose final state is our desired edit representation. In this work, we use a biLSTM.

---

[3] EXPANDR corresponds to the APPLYCONSTR action in the original model of Yin & Neubig (2018). There is also a REDUCE action which marks the end of expanding a non-terminal with non-deterministic number of child nodes. See Yin & Neubig (2018) for details.

**Graph Encoding of Edits**    As in the graph-to-tree neural editor, we represent $x_-$ and $x_+$ as trees $T(x_-)$ and $T(x_+)$. We combine these trees into a graph representation $G(x_- \to x_+)$ by merging both trees into one graph, using "Removed", "Added" and "Replaced" edges. To connect the two trees, we compute the same alignment as in the sequence case, connecting leaves that are the same and each replaced leaf to its replacement. We also propagate this information up in the trees, *i.e.*, two inner nodes are connected by "=" edges if all their descendants are connected by "=" edges. This is illustrated in Figure 3(b). Finally, we also use the same "+" / "-" / "↔" / "=" tags for the initial node representation, computing it as the concatenation of the string label (*i.e.* token or nonterminal name) and the embedding of the tag. To obtain an edit representation, we use a GGNN unrolled for a fixed number of timesteps and again use the weighted averaging strategy of Gilmer et al. (2017).

## 4    EVALUATION

Evaluating an unsupervised representation learning method is challenging, especially for a newly defined task. Here, we aim to evaluate the quality of the learned edit representations with a series of qualitative and quantitative metrics on natural language and source code.

### 4.1    DATASETS AND CONFIGURATION

**Natural Language Edits**    We use the **WikiAtomicEdits** (Faruqui et al., 2018) dataset of pairs of short edits on Wikipedia articles. We sampled $1040K$ edits from the English *insertion* portion of the dataset and split the samples into $1000K/20K/20K$ train-valid-test sets.

**Source Code Edits**    To obtain a dataset for source code, we clone a set of 54 C# projects on GitHub and collected a **GitHubEdits** dataset (see Appendix A for more information). We selected all changes in the projects that are no more than 3 lines long and whose surrounding 3 lines of code before and after the edited lines have *not* been changed, ensuring that the edits are separate and short. We then parsed the two versions of the source code and take as $x_-$ and $x_+$ the code that belongs to the top-most AST node that contains the edited lines. Finally, we remove trivial changes such variable renaming, changes within comments or formatting changes. Overall, this yields 111 724 edit samples. For each edit we run a simple C# analysis to detect all variables and normalize variable names such that each unique variable within $x_-$ and $x_+$ has a unique normalized name V0, V1, *etc*. This step is necessary to avoid the sparsity of data induced by the variety of different identifier naming schemes. We split the dataset into 91,372 / 10,176 / 10,176 samples as train/valid/test sets.

Additionally, we introduce a labeled dataset of source code edits by using C# "fixers". Fixers are small tools built on top of the C# compiler, used to perform common refactoring and modernization tasks (*e.g.*, using new syntactic sugar). We selected 16 of these fixers and ran them on 6 C# projects to generate a small **C#Fixers** dataset of 2,878 edit pairs with known semantics. We present descriptions and examples of each fixer in Appendix A.

**Configuration**    Throughout the evaluation we use a fixed size of 512 for edit representations. The size of word embeddings and hidden states of encoding LSTMs is 128. The dimensionality of the decoding LSTM is set to 256. Details of model configuration can be found in Sect. A.

When generating the target $x_+$, our neural editor model can optionally take as input the context of the original input $x_-$ (e.g., the preceding and succeeding code segments surrounding $x_-$), whose information could be useful for predicting $x_+$. For example, in source code edits the updated code snippet $x_+$ may reuse variables defined in the preceding snippet. In our code experiments, we use a standard bidirectional LSTM network to encode the tokenized 3 lines of code before and after $x_-$ as context. The encoded context is used to initialize the decoder, and as an additional source for the pointer network to copy tokens from.

### 4.2    QUALITY OF EDIT REPRESENTATIONS

First, we study the ability of our models to encode edits in a semantically meaningful way.

**Visualizing Edits on Fixers Data** In a first experiment, we train our sequential neural editor model on our GitHubEdits data and then compute representations for the edits generated by the C# fixers. A t-SNE visualization (Maaten & Hinton, 2008) of the encodings is shown in Figure 4. For this visualization, we randomly selected 100 examples from the edits of each fixer (if that fixer has more than 100 samples) and discarded fixer categories with less than 40 examples. Readers are referred to Appendix A for detailed descriptions of each fixer category. We find that our model produces dense clusters for simple or distinctive code edits, *e.g.* fixer RCS1089 (using the ++ or -- unary operators instead of a binary operator (*e.g.*, i = i + 1 → i++), and fixer CA2007 (adding .ConfigureAwait(false) for await statements). We also analyzed cases where (1) the edit examples from the same fixer

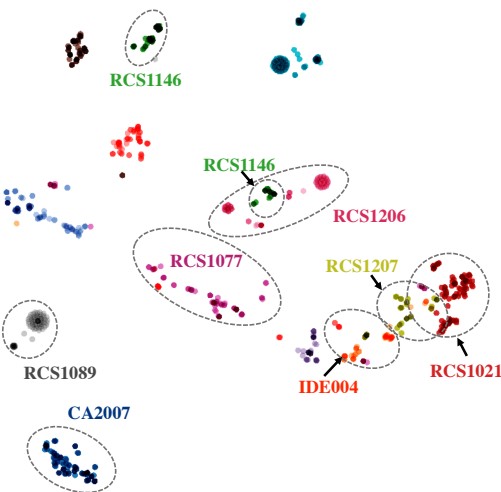

Figure 4: t-SNE visualization of edits from 13 C# fixers, where point color indicates the fixer. Labels indicate the id of the fixer, see main text.

are scattered, or (2) the clusters of different fixers overlap with each other. For example, the fixer RCS1077 covers 12 different aspects of optimizing LINQ method calls (*e.g.*, type casting, counting, etc.), and hence its edits are scattered. On the other hand, fixers RCS1146 and RCS1206 yield overlapping clusters, as both fixers change code to use the ?. operator. Fixers RCS1207 (change a lambda to a method group, *e.g.* foo(x=>bar(x)) → foo(bar)) and RCS1021 (simplify lambda expressions, *e.g.* foo(x=>{return 4;}) → foo(x=>4)) are similar, as both inline lambda expressions in two different ways. Analysis yields that the representation is highly dependent on surface tokens. For instance, IDE004 (removing redundant type casts, *e.g.* (int)2 → 2) and RCS1207 (removing explicit argument lists) yield overlapping clusters, as both involve deleting identifiers wrapped by parentheses.

**Human Evaluation on Encoding Natural Language WikiAtomicEdits** In a second experiment, we test how well neighborhoods in edit representation space correspond to semantic similarity. We computed the five nearest neighbors of 200 randomly sampled seed edits from our training set, using both our trained sequence-to-sequence editing model with sequential edit encoder, as well as a simple bag-of-words baseline based on TF-IDF scores. We then rated the quality of the retrieved neighbors on a scale of 0 ("unrelated edit"), 1 ("similar edit") and 2 ("semantically or syntactically same edit"). Details of the annotation schema is included in Sect. E. We show the (normalized) discounted cumulative gain (DCG, Manning et al. (2008)) for the two models at the top of Tab. 1 (higher is better). The relevance scores indicate that our neural model clearly outperforms the simplistic baseline. Tab. 1 also presents two example edits with their nearest neighbors. Example 1 shows that the neural edit models succeeded in representing syntactically and semantically similar edits, while the bag-of-words baseline relies purely on surface token overlap. Interestingly, we also observed that the edit representations learned by the neural editing model on WikiAtomicEdits are somewhat sensitive to position, *i.e.* the position of the inserted tokens in both the seed edit and the nearest neighbors is similar. This is illustrated in Example 2, where the second (*"senegalese striker"*) and the third (*"republican incumbent"*) nearest neighbors returned by the neural model have similar editing positions as the seed edit, while they are semantically diverse.

## 4.3 EDIT ENCODER PERFORMANCE

To evaluate the performance of our two edit encoders discussed in Sect. 3.2 and disentangle it from the choice of neural editor, we train various combinations of our neural editor model and manually evaluate the quality of the edit representation. More specifically, we trained our neural editor models on GitHubEdits and randomly sampled 200 seed edits and computed their 3 nearest neighbors using each end-to-end model. We then rated the resulting groups using the same 0-2 scale as above. The resulting relevance scores are shown in Tab. 2.

| | Bag of Words Model | Seq2Seq – Seq Edit Encoder |
|---|---|---|
| DCG/NDCG@5 | 9.3 / 67.3% | 13.5 / 90.3% |
| DCG@5 (by edit size) | 1: 14.7  2-3: 10.8  >3: 5.4 | 1: 16.2  2-3: 12.9  >3: 12.4 |

**Example 1** ▶*daniel james nava ( born february 22 , 1983 ) is an american professional baseball outfielder*◀ nava is only the fourth player in mlb history to hit a grand slam in his first major league at bat and the second to do it on the first pitch .

| | | |
|---|---|---|
| NN-1 | he batted .302 with 73 steals , and received a september call - up to the major leagues ▶*as an outfielder*◀ . | ▶*arthur ray briles ( born december 3 , 1955 ) is a former american football coach and*◀ his most recent head coaching position was at baylor university , a position he held from the 2008 season through the 2015 season . |
| NN-2 | he played ▶*as an outfielder*◀ for the hanshin tigers . | ▶*jonathan david disalvatore ( born march 30 , 1981 ) is a professional ice hockey*◀ he was selected by the san jose sharks in the 4th round ( 104th overall ) of the 2000 nhl entry draft . |
| NN-3 | in 2012 , his senior at oak mountain , dahl had a .412 batting average , 34 runs batted in ( rbis ) , and 18 stolen bases ▶*as an outfielder .*◀ | ▶*professor paul talalay ( born march 31 , 1923 ) is the john jacob abel*◀ distinguished service professor of pharmacology and director of the laboratory for molecular sciences at johns hopkins school of medicine in baltimore . |

**Example 2** she , along with her follow artist carolyn mase studied with ▶*impressionist landscape painter*◀ john henry twachtman at the art students league of new york .

| | | |
|---|---|---|
| NN-1 | his brother was draughtsman william daniell and his uncle was ▶*landscape painter*◀ thomas daniell . | the first painting was a portrait of a young girl , emerantia van beresteyn , the sister of ▶*the landscape painter*◀ nicolaes van beresteyn , the later founder of half of this hofje . |
| NN-2 | william james linton ( december 7 , 1812 - december 29 , 1897 ) was an english - born american wood engraver , ▶*landscape painter ,*◀ political reformer and author of memoirs , novels , poetry and non-fiction . | he was the club 's top scorer with 22 goals in all competitions , one more than ▶*senegalese striker*◀ lamine diarra , who left the club at the end of the season . |
| NN-3 | early on , hopper modeled his style after chase and french ▶*impressionist*◀ masters douard manet and edgar degas . | caforio ” aggressively attacked ” his opponent , ▶*republican incumbent*◀ steve knight , for his delayed response to the leak . |

Table 1: Natural language human evaluation results and 3 nearest neighbors. ▶*Inserted text*◀ marked. **Example 1** neural editing model returns syntactically and semantically similar edits. **Example 2** Neural edit representations are sensitive to position.

Table 2: Relevance scores of human evaluation on GitHubEdits data. Acc.@1 denotes the ratio that the 1-nearest neighbor has a score 2.

| Model | DCG@3 | NDCG@3 (%) | Acc.@1 (%) |
|---|---|---|---|
| BoW | 7.77 | 75.99 | 58.46 |
| Seq2Seq – Seq Edit Encoder | 10.09 | 90.05 | 75.90 |
| Graph2Tree – Seq Edit Encoder | **10.56** | **91.40** | **79.49** |
| Graph2Tree – Graph Edit Encoder | 9.44 | 86.20 | 72.31 |

Comparing the sequential edit encoders trained with Seq2Seq and Graph2Tree editors, we found that the edit encoder trained with the Graph2Tree objective performs better. We hypothesize that this is because the Graph2Tree editor better captures structural-level information about an edit. For instance, Example 1 in Tab. 3 removes explicit type casting. The Seq2Seq editor has difficulty distinguishing this type of edit, confusing it with changes of lambda expressions to method groups (1st and 2nd nearest neighbors) since both two types of edits involve removing paired parentheses.

Surprisingly, we found that the graph-based edit encoder does not outperform the sequence-based encoder. However, we observe that the graph edit encoder in many cases tends to better capture high-level and abstract structural edit patterns. Example 2 in Tab. 3 showcases a seed edit that swaps two consecutive declarations, which corresponds to swapping the intermediate Expression nodes representing each statement on the underlying AST. In this case, the graph edit encoder is capable of grouping semantically similar edits, while it seems to be more difficult for the sequential encoder

Table 3: Two example source code edits and their nearest neighbors based on the edit representations computed by each model.

| Example 1 | Example 2 |
|---|---|
| $\boldsymbol{x}_-$: `V0.SendSelectSoundRequest((int)V1);`
$\boldsymbol{x}_+$: `V0.SendSelectSoundRequest(V1);` | $\boldsymbol{x}_-$: `string V0; string V1;`
$\boldsymbol{x}_+$: `string V1; string V0;` |

| Seq2Seq – Seq Edit Encoder | Seq2Seq – Seq Edit Encoder |
|---|---|
| ▶ $\boldsymbol{x}_-$: `V0.Debug(() => LITERAL);`
$\boldsymbol{x}_+$: `V0.Debug(LITERAL);` | ▶ $\boldsymbol{x}_-$: `RetryConfig V0; string V1;`
$\boldsymbol{x}_+$: `string V1; RetryConfig V0;` |
| ▶ $\boldsymbol{x}_-$: `V0.Debug(() => LITERAL);`
$\boldsymbol{x}_+$: `V0.Debug(LITERAL);` | ▶ $\boldsymbol{x}_-$: `string[] V0; string[] V1; int V2;`
$\boldsymbol{x}_+$: `int V2; string[] V0; string[] V1;` |
| ▶ $\boldsymbol{x}_-$: `V0.WriteCompressedInteger((uint)V1);`
$\boldsymbol{x}_+$: `V0.WriteCompressedInteger(V1);` | ▶ $\boldsymbol{x}_-$: `Type V0= null; BindingFlags V1= 0;`
$\boldsymbol{x}_+$: `BindingFlags V1= 0; Type V0= null;` |

| Graph2Tree – Seq Edit Encoder | Graph2Tree – Seq Edit Encoder |
|---|---|
| ▶ $\boldsymbol{x}_-$: `V0.WriteCompressedInteger((uint)V1);`
$\boldsymbol{x}_+$: `V0.WriteCompressedInteger(V1);` | ▶ $\boldsymbol{x}_-$: `RetryConfig V0; string V1;`
$\boldsymbol{x}_+$: `string V1; RetryConfig V0;` |
| ▶ $\boldsymbol{x}_-$: `V0.WriteCompressedInteger((uint)V1);`
$\boldsymbol{x}_+$: `V0.WriteCompressedInteger(V1);` | ▶ $\boldsymbol{x}_-$: `string[] V0; string[] V1; int V2;`
$\boldsymbol{x}_+$: `int V2; string[] V0; string[] V1;` |
| ▶ $\boldsymbol{x}_-$: `V0.WriteCompressedInteger((uint)V1);`
$\boldsymbol{x}_+$: `V0.WriteCompressedInteger(V1);` | ▶ $\boldsymbol{x}_-$: `int V0 = V1; int V2 = V3;`
$\boldsymbol{x}_+$: `int V2 = V3; int V0 = V1;` |

| Graph2Tree – Graph Edit Encoder | Graph2Tree – Graph Edit Encoder |
|---|---|
| ▶ $\boldsymbol{x}_-$: `V0.UpdateLastRead(this.V1);`
$\boldsymbol{x}_+$: `V0.UpdateLastRead(V1);` | ▶ $\boldsymbol{x}_-$: `RetryConfig V0; string V1;`
$\boldsymbol{x}_+$: `string V1; RetryConfig V0;` |
| ▶ $\boldsymbol{x}_-$: `V0.UpdateLastWrite(this.V1);`
$\boldsymbol{x}_+$: `V0.UpdateLastWrite(V1);` | ▶ $\boldsymbol{x}_-$: `int V0 = V1; int V2 = V3;`
$\boldsymbol{x}_+$: `int V2 = V3; int V0 = V1;` |
| ▶ $\boldsymbol{x}_-$: `V0.Append(this.V1);`
$\boldsymbol{x}_+$: `V0.Append(V1);` | ▶ $\boldsymbol{x}_-$: `double V0= -1; double V1= -1;`
$\boldsymbol{x}_+$: `double V1= -1; double V0= -1;` |

Table 4: Test performance of different neural editors.

| Model | Acc.@1 (%) | Recall@5 (%) | PPL per token |
|---|---|---|---|
| GitHubEdits | | | |
|    Seq2Seq – Bag-of-Edits Encoder | 44.05 | 54.97 | 1.4808 |
|    Seq2Seq – Seq Edit Encoder | **59.63** | **65.46** | **1.2792** |
|    Graph2Tree – Bag-of-Edits Encoder | 40.66 | 49.42 | 1.5058 |
|    Graph2Tree – Seq Edit Encoder | 57.49 | 62.94 | 1.3043 |
|    Graph2Tree – Graph Edit Encoder | 48.05 | 56.51 | 1.3712 |
| WikiAtomicEdits | | | |
|    Seq2Seq – Bag-of-Edits Encoder | 23.73 | 43.47 | 1.3730 |
|    Seq2Seq – Seq Edit Encoder | **72.94** | **76.53** | **1.0527** |

encoder to capture the edit pattern. On the other hand, we found that the graph edit encoder often fails to capture simpler, lexical level edits (*e.g.*, Example 1). This might suggest that terminal node information is not effectively propagated, an interesting issue worth future investigation.

## 4.4 PRECISION OF NEURAL EDITORS

Finally, we evaluate the performance of our end-to-end system by predicting the edited input $\boldsymbol{x}_+$ given $\boldsymbol{x}_-$ and the edit representation. We are interested in answering two research questions: *First*, how well can our neural editors generate $\boldsymbol{x}_+$ given the gold-standard edit representation $f_\Delta(\boldsymbol{x}_-, \boldsymbol{x}_+)$? *Second*, and perhaps more interestingly, can we use the representation of a similar edit $f_\Delta(\boldsymbol{x}'_-, \boldsymbol{x}'_+)$ to generate $\boldsymbol{x}_+$ by *applying* that edit to $\boldsymbol{x}_-$ (*i.e.* $\hat{\boldsymbol{x}}_+ = \alpha(\boldsymbol{x}_-, f_\Delta(\boldsymbol{x}'_-, \boldsymbol{x}'_+)))$?

To answer the first question, we trained our neural editor models on the WikiAtomicEdits and the GitHubEdits dataset, and evaluate the performance of encoding and applying edits on test sets. For completeness, we also evaluated the performance of our neural editor models with a simple "Bag-of-Edits" edit encoding scheme, where $f_\Delta(\boldsymbol{x}_-, \boldsymbol{x}_+)$ is modeled as the concatenation of two vectors, each representing the sum of the embeddings of added and deleted tokens in the edit, respectively. This edit encoding method is reminiscent of the model used in Guu et al. (2017) for solving a different task of language modeling by marginalizing over latent edits, which we will elaborate in Sect. 5. Tab. 4

Table 5: Transfer learning results on C# fixers data, averaged across all fixer categories.

| Model | Acc.(%) | Acc.$^{*}$(%) | Recall@5(%) | Recall@5$^{*}$(%) |
|---|---|---|---|---|
| Seq2Seq – Seq Edit Encoder | 38.35 | **77.67** | 41.50 | **83.84** |
| Graph2Tree – Seq Edit Encoder | **49.21** | 77.30 | **51.93** | 81.77 |
| Baselines (no edit encoding) | | | | |
| Seq2Seq w/o Edit Encoder | 7.07 | — | 14.29 | — |
| Graph2Tree w/o Edit Encoder | 8.81 | — | 11.90 | — |

$^{*}$: upper-bound performance of predicting $\boldsymbol{x}_+$ using the gold-standard edit representations.

lists the evaluation results. With our proposed sequence- and graph-based edit encoders, our neural editor models achieve reasonable end-to-end performance, surpassing systems using bag-of-edits representations. This is because many edits are *context-sensitive* and *position-sensitive*, requiring edit representation models that go beyond the bag-of-edits scheme to capture those effects (more analysis is included in Appendix B). Interestingly, on the GitHubEdits dataset, we find that the Seq2Seq editor with sequential edit encoder registers the best performance. However, it should be noted that in this set of experiments, we encode the gold-standard edit $f_\Delta(\boldsymbol{x}_-, \boldsymbol{x}_+)$ to predict $\boldsymbol{x}_+$. As we will show later, better performance with the gold-standard edit does *not* necessarily imply better (more generalizable) edit representation. Nevertheless, we hypothesize that the higher accuracy of the Seq2Seq edit is due to the fact that a significant proportion of edits in this dataset is small and primarily syntactically simple. Indeed we find that 69% of test examples have a token-level edit distance of less than 5.

To answer the second question, we use the trained neural editors from the previous experiment, and test their performance in a "one-shot" transfer learning scenario. Specifically, we use our high-quality C#Fixers dataset, and for each fixer category $\mathcal{F}$ of semantically similar edits, we randomly select a seed edit $\{\boldsymbol{x}'_- \to \boldsymbol{x}'_+\} \in \mathcal{F}$, and use its edit representation $f_\Delta(\boldsymbol{x}'_-, \boldsymbol{x}'_+)$ to predict the updated code for *all* examples in $\mathcal{F}$, *i.e.*, we have $\hat{\boldsymbol{x}}_+ = \alpha(\boldsymbol{x}_-, f_\Delta(\boldsymbol{x}'_-, \boldsymbol{x}'_+)), \forall \{\boldsymbol{x}_- \to \boldsymbol{x}_+\} \in \mathcal{F}$. This task is highly non-trivial, since a fixer category could contain more than hundreds of edit examples collected from different C# projects. Therefore, it requires the edit representations to generalize and transfer well, while being invariant of local lexical information like specific method names. To make the experimental evaluation more robust to noise, for each fixer category $\mathcal{F}$, we randomly sample 10 seed edit pairs $\{\boldsymbol{x}'_- \to \boldsymbol{x}'_+\}$, compute their edit representations and use them to predict the edited version of the examples in $\mathcal{F}$ and evaluate accuracy of predicting the exact final version. We then report the best score among the 10 seed representations as the performance metric on $\mathcal{F}$.

Tab. 5 summarizes the results and also reports the upper bound performance when using the gold-standard edit representation $f_\Delta(\boldsymbol{x}_-, \boldsymbol{x}_+)$ to predict $\boldsymbol{x}_+$, and an approximation of the "lower bound" accuracies using pre-trained Seq2Seq and Graph2Tree models without edit encoders. We found that our neural Graph2Tree editor with the sequential edit encoder significantly outperforms the Seq2Seq editor, even though Seq2Seq performs better when using gold-standard edit representations. This suggest that the edit representations learned with the Graph2Tree model generalize better, especially for edits discussed in Sect. 4.2 that involve syntactic variations like `RCS1021` (lambda expression simplification, 7.8% *vs.* 30.7% for Seq2Seq and Graph2Tree), and `RCS1207` (change lambdas to method groups, 7.1% *vs.* 26.2%). Interestingly, we also observe that Seq2Seq outperforms the Graph2Tree model for edits with trivial surface edit sequences, where the Graph2Tree model does not have a clear advantage. For example, on `RCS1015` (use `nameof` operator, *e.g.* `Exception("x") → Exception(nameof(x))`), the accuracies for Seq2Seq and Graph2Tree are 40.0% (14/35) and 28.6% (10/35), resp. We include more analysis of the results in Appendix C.

## 5 RELATED WORK

Edits have recently been considered in NLP, as they represent interesting linguistic phenomena in language modeling and discourse (Faruqui et al., 2018; Yang et al., 2017a). Specifically, Guu et al. (2017) present a generative model of natural language sentences via editing prototypes. Our work shares with Guu et al. (2017) in that (1) the posterior edit encoding model in Guu et al. (2017) is similar to our baseline "bag-of-edits" encoder in Sec. 4.4, and (2) the sequence-to-sequence sentence generation model given the prototype and edit representation is reminiscent of our Seq2Seq editor. In contrast, our work directly focuses on discriminative learning of representing edits and applying the learned edits for both sequential (NL) and structured (code) data. Another similar line of research

is "retrieve-and-edit" models for text generation (Hashimoto et al., 2018), where given an input data $x$, the target prediction $y$ is generated by editing a similar target $y'$ that is retrieved based on the similarity between its source $x'$ and the input $x$. While these models typically require an "editor" component to generate the output by exploiting the difference between similar inputs, they usually use the simpler bag-of-edits representations (Wu et al., 2019), or implicitly capture it via end-to-end neural networks (Contractor et al., 2018). To our best knowledge, there is not any related work that classifies or otherwise explicitly represents the differences over similar input, with the exception of differential recurrent neural networks used for action recognition in videos (Veeriah et al., 2015; Zhuang et al., 2018). This is a substantially different task, as the data includes a temporal component as well.

Source code edits are a widely studied artifact. Specialized software, such as git, is widely used to store source code revision histories. Nguyen et al. (2013) studied the repetitiveness of source code changes by identifying identical types of changes using a deterministic differencing tool. In contrast, we employ on a neural network to cluster *similar* changes together. Rolim et al. (2017) use such clusters to synthesize small programs that perform the edit. The approach is based on Rolim et al. (2018) extract manually designed syntactic features from code and cluster over multiple changes to find repeatable edit rules. Similarly, Paletov et al. (2018) extract syntactic features specifically targeting edits in cryptography API protocols. In this work, we try to avoid hand-designed features and allow a neural network to learn the relevant aspects of a change by directly giving as input the original and final version of a changed code snippet.

Modeling tree generation with machine learning is an old problem that has been widely studied in NLP. Starting with Maddison & Tarlow (2014), code generation has also been considered as a tree generation problem. Close to our work is the decoder of Yin & Neubig (2017) which we use as the basis of our decoder. The work of Chen et al. (2018) is also related, since it provides a tree-to-tree model, but focuses on learning a single translation tasks and cannot be used directly to represent multiple types of edits. Both Yin & Neubig (2017) and Chen et al. (2018) have copying mechanism for single tokens, but our subtree copying mechanism is novel.

Autoencoders (see Goodfellow et al. (2016) for an overview) have a long history in machine learning. Variational autoencoders (Kingma & Welling, 2013) are similar to autoencoders but instead of focusing on the learned representation, they aim to create accurate generative probabilistic models. Most (variational) autoencoders focus on encoding images but there have been works that autoencode sequences, such as text (Dai & Le, 2015; Bowman et al., 2015; Yang et al., 2017b) and graphs (Simonovsky & Komodakis, 2018; Liu et al., 2018). Conditional variational autoencoders (Sohn et al., 2015) have a related form to our model (with the exception of the KL term), but are studied as generative models, whereas we are primarily interested in the edit representation.

## 6 DISCUSSION & CONCLUSIONS

In this work, we presented the problem of learning distributed representation of edits. We believe that the dataset of edits is highly relevant and should be studied in more detail. While we have presented a set of initial models and metrics on the problem and obtained some first promising results, further development in both of these areas is needed. We hope that our work inspires others to work on this interesting problem in the future.

### ACKNOWLEDGMENTS

We would like to thank Rachel Free for her insightful comments and suggestions.

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

## A  DATASETS AND CONFIGURATION

**WikiAtomicEdits**   We randomly sampled $1040K$ insertion examples from the English portion of WikiAtomicEdits (Faruqui et al., 2018) dataset, with a train, development and test splits of $1000K$, $20K$ and $20K$.

**GitHubEdits**   We cloned the top 54 C# GitHub repositories based on their popularity (Tab. 8). For each commit in the `master` branch, we collect the previous and updated versions of the source code, and extract all consecutive lines of edits that are smaller than three lines, and with at least three preceding and successive lines that have not been changed. We then filter trivial changes such as variable and identifier renaming, and changes happened within comments. We also limit the number of tokens for each edit to be smaller than 100, and down-sample edits whose frequency is larger than 30. Finally, we split the dataset by commit ids, ensuring that there are no edits in the training and testing (development) sets coming from the same commit. Tab. 6 lists some statistics of the dataset.

Table 6: Statistics of the GitHubEdits Dataset

| | |
|---|---|
| Average Num. Tokens in $x_-$ | 16.4 |
| Average Num. Tokens in $x_+$ | 17.0 |
| Average Edit Distance | 5.0 |
| Average size of AST for $T(x_-)$ | 28.5 |
| Average size of AST for $T(x_+)$ | 29.4 |

**C#Fixers**   We selected 16 C# fixers from Roslyn[4] and Roslynator[5], and ran them on 6 C# projects to generate a small, high-quality C# fixers dataset of $2\,878$ edit pairs with known semantics. Table 7 lists the detailed descriptions for each fixer category. And more information can be found at `https://github.com/JosefPihrt/Roslynator/blob/master/src/Analyzers/README.md`.

**Network Configuration**   Throughout the experiments, we use a fixed edit representation size of 512. The dimensionality of word embedding, the hidden states of the encoder LSTMs, as well as the gated graph neural network is 128, while the decoder LSTM uses a larger hidden size of 256. For the graph-based edit encoder, we used a two-layer graph neural network, with 5 information propagation steps at each layer. During training, we performed early stopping, and choose the best model based on perplexity scores on development set. During testing, we decode a target element $x_+$ using a beam size of 5.

---

[4] `http://roslyn.io`
[5] `https://github.com/JosefPihrt/Roslynator`

Table 7: Descriptions of fixer categories in C#Fixers dataset

| Fixer ID | Description | Num. Edits | Example |
|---|---|---|---|
| CA2007 | apply `.ConfigureAwait(false)` to `await` statements | 1051 | $x_-$: `await Console.WriteAsync()`
$x_+$: `await Console.WriteAsync()`
`                    .ConfigureAwait(false)` |
| IDE0004 | Cast is redundant | 53 | $x_-$: `var x = 1; var b = (int)x;`
$x_+$: `var x = 1; var b = x;` |
| RCS1015 | Use `nameof` operator | 35 | $x_-$: `Exception("parameter");`
$x_+$: `Exception(nameof(parameter));` |
| RCS1021 | Simplify lambda expression | 411 | $x_-$: `var x = items.Select(f =>`
`{`
`    return f.ToString();`
`});`
$x_+$: `var x = items.Select(`
`     f => f.ToString());` |
| RCS1032 | Remove redundant parentheses | 24 | $x_-$: `if ((x)) {}`
$x_-$: `if (x) {}` |
| RCS1058 | Use compound assignment | 43 | $x_-$: `i = i + 2;`
$x_+$: `i += 2;` |
| RCS1077 | Optimize LINQ method call | 200 | $x_-$: `items.Where(f => Foo(f)).Any();`
$x_+$: `items.Any(f => Foo(f));` |
| RCS1089 | Use `--`/`++` operator instead of assignment | 75 | $x_-$: `i = i + 1;`
$x_+$: `i += 1;` |
| RCS1097 | Remove redundant `ToString` call | 20 | $x_-$: `var x = s.ToString();`
$x_+$: `var x = s;` |
| RCS1118 | Mark local variable as const | 477 | $x_-$: `string s = "a";`
`string s2 = s + "b";`
$x_+$: `const string s = "a";`
`string s2 = s + "b";` |
| RCS1123 | Add parentheses according to operator precedence | 109 | $x_-$: `if (x || y && z) {}`
$x_+$: `if (x || (y && z) ) {}` |
| RCS1146 | Use conditional access | 71 | $x_-$: `x != null && x.StartsWith("a");`
$x_+$: `x?.StartsWith("a");` |
| RCS1197 | Optimize call of `StringBuilder`'s `Append`/`AppendLine` | 95 | $x_-$: `sb.Append(s + "x");`
$x_+$: `sb.Append(s).Append("x");` |
| RCS1202 | Avoid `NullReferenceException` | 56 | $x_-$: `items.First().ToString();`
$x_+$: `items?.First().ToString();` |
| RCS1206 | Use conditional access instead of conditional expression | 116 | $x_-$: `int i = (x != null) ?`
`     x.Value.GetHashCode() : 0;`
$x_+$: `int i = x?.GetHashCode() ?? 0;` |
| RCS1207 | Use method group instead of anonymous function | 42 | $x_-$: `items.Select(f => Foo(f));`
$x_+$: `items.Select(Foo);` |

Table 8: Our C# GitHub dataset projects

| Name | GitHub Id | Description |
|---|---|---|
| acat | intel/acat | Assistive Context-Aware Toolkit |
| akka.net | akka/akka.net | Distributed Actors |
| aspnetboilerplate | aspnetboilerplate/aspnetboilerplate | ASP.NET boilerplate |
| AutoMapper | AutoMaper/AutoMapper | Object-Object Mapper |
| BotBuilder | Microsoft/BotBuilder | Bot Framework |
| CefSharp | cefsharp/CefSharp | Chromium Embedded Framework Bindings |
| choco | chocolatey/choco | package mananger |
| cli | dotnet/cli | .NET CLI Tools |
| CodeHub | CodeHubApp/CodeHub | iOS application |
| coreclr | dotnet/coreclr | .NET Framework |
| corefx | dotnet/corefx | .NET FOundational Libraries |
| dapper | StackExchange/Dapper | Object Mapper |
| dnSpy | 0xd4d/dnSpy | .NET debugger and assembly editor |
| duplicati | duplicati/duplicati | Encrypted Cloud Backups |
| EntityFramework | aspnet/EntityFramework | Object-Relational Mapper |
| EntityFrameworkCore | aspnet/EntityFrameworkCore | Object-Relational Mapper – Core |
| FluentValidation | JeremySkinner/FluentValidation | Validation Rules |
| framework | accord-net/framework | ML, CV Framework |
| GVFS | Microsoft/VFSForGit | Git Virual File System |
| Hangfire | HangfireIO/Hangfire | Background job library |
| ILSpy | icsharpcode/ILSpy | Decompiler |
| JavaScriptServices | aspnet/JavaScriptServices | ASP.NET JS Services |
| MahApps.Metro | MahApps/MahApps.Metro | WPF Framework |
| MaterialDesignInXamlToolkit | MaterialDesignInXamlToolkit/ MaterialDesignInXamlToolkit | Design XAML & WPF |
| mono | mono/mono | .NET implementation |
| monodevelop | mono/monodevelop | IDE |
| MonoGame | MonoGame/MonoGame | Game Framework |
| msbuild | Microsoft/msbuild | Build Tool |
| Mvc | aspnet/Mvc | MVC Framework |
| Nancy | NancyFx/Nancy | HTTP based services |
| Newtonsoft.Json | JamesNK/Newtonsoft.Json | JSON framework |
| NLog | NLog/NLog | Loggin for .NET |
| OpenLiveWriter | OpenLiveWriter/ OpenLiveWriter | Text editor |
| OpenRA | OpenRA/OpenRA | Strategy Game Engine |
| Opserver | opserver/Opserver | Monitoring System |
| orleans | dotnet/orleans | Distributed Virtual Actors |
| PowerShell | PowerShell/PowerShell | Command Line |
| Psychson | brandonlw/Psychson | Firmware |
| PushSharp | Redth/PushSharp | Push Notifications |
| ravendb | ravendb/ravendb | Database |
| ReactiveUI | reactiveui/ReactiveUI | Reactive MVC Framework |
| RestSharp | restsharp/RestSharp | HTTP/REST Client |
| roslyn | dotnet/roslyn | .NET Compiler |
| Rx.NET | dotnet/reactive | Reactive extensions. |
| ServiceStack | ServiceStack/ServiceStack | Web Service Framework |
| shadowsocks-windows | shadowsocks/ shadowsocks-windows | Cryptography |
| ShareX | ShareX/ShareX | Screen Recorder |
| SignalR | SignalR/SignalR | Real-time web framework |
| Sonarr | Sonarr/Sonarr | PVR |
| SpaceEngineers | KeenSoftwareHouse/ SpaceEngineers | Game |
| SparkleShare | hbons/SparkleShare | File Sharing |
| StackExchange.Redis | StackExchange/ StackExchange.Redis | Redis Client |
| WaveFunctionCollapse | mxgmn/ WaveFunctionCollapse | Bitmap/tilemap Generator |
| Wox | Wox-launcher/Wox | Launcher |

## B  CLUSTERING EXPERIMENTS

To qualitatively evaluate the quality of the learned edit representations. We use the models trained on the WikiAtomicEdits and GitHubEdits datasets to cluster natural language and code edits. We run K-Means clustering algorithm on $0.5$ million sampled edits from WikiAtomicEdits, and all $90K$ code edits from GitHubEdits, producing $50\,000$ and $20\,000$ clusters for each dataset.

Tab. 9 and Tab. 10 list some example clusters on WikiAtomicEdits and GitHub datasets, respectively. Due to the size of clusters, we omit out-liners and present distinctive examples from each cluster. On the WikiAtomicEdits dataset, we found clusters whose examples are semantically and syntactically similar. More interestingly, on the source code data, we find representative clusters that relate to idiomatic patterns and best practices of programming. The clustering results produced by our model would be useful for programming synthesis toolkits to generate interpretable code refractory rules, which we leave as interesting future work.

Finally, we remark that the clustering results indicate that the encoding of edits is *context-sensitive* and *position-sensitive* for both natural language and source code data. For instance, the WikiAtomicEdits examples we present in Tab. 9 clearly indicate that semantically similar insertions also share similar editing positions. This is even more visible in code edits (Tab. 10). For instance, in the first example in Tab. 10, `Equal()` can be changed to `Empty()` only in the `Assert` namespace (i.e., the context). These examples demonstrate that it is important for an edit encoder to capture the contextual and positional information in edits, a property that cannot be captured by simple "bag-of-edits" edit representation methods.

Table 9: Example clusters on WikiAtomicEdits data using representations learned by a neural Seq2Seq editor with sequential edit encoder

Description  Add a person's middle name

1. isaiah ►*marcus*◄ rankin ( born 22 may 1978 in london ) is an english professional footballer currently playing for stevenage borough .
2. audrey ►*kathleen*◄ brown ( born 24 may , 1913 ) is a british athlete who competed mainly in the 100 metres .
3. alice ►*edith*◄ rumph was a painter , etcher , and teacher .
4. mark ►*larry*◄ taufua is an australian professional rugby league player .
5. monique ►*edith*◄ lamoureux ( born july 3 , 1989 ) is an american ice hockey player .

Description  Add a parenthetical expression *also ... as* to modify the subject

1. mid-state regional airport ►*, also known as mid-state airport ,*◄ is a small airport on in rush township , centre county in pennsylvania in the united states .
2. islamic culture ►*, also known as saracenic culture ,*◄ is a term primarily used in secular academia to describe the cultural practices common to historically islamic peoples .
3. birds of prey ►*, also known as raptors ,*◄ are birds that hunt for food primarily via flight , using their keen senses , especially vision .
4. tetyana styazhkina ►*, also written as tetyana stiajkina ,*◄ ( ; born april 10 , 1977 ) is a ukrainian cycle racer who rides for the chirio forno d'asolo team .
5. acid jazz ►*, also known as club jazz ,*◄ is a musical genre that combines elements of jazz , soul , funk , disco and hip hop .

Description  Specify location using a prepositional phrase.

1. the douro fully enters portuguese territory just after the confluence with the gueda river ; once the douro enters portugal , major population centres are less frequent ►*along the river*◄ .
2. mochou lake and mochou lake park are located at 35 hanzhongmen da jie in the jianye district of nanjing , china ►*, west to qinhuai river*◄ .
3. reiner gamma is an albedo feature that is located on the oceanus procellarum , to the west of the reiner crater ►*on the moon*◄ .
4. she made a brief return to the screen in " parrish " ( 1961 ) , playing the supporting role of mother which received little attention ►*by the press*◄ .
5. he was involved in a few storylines , including one where he broke his toe and had a heart attack after he was pushed by a mugger ►*in the market*◄ .

Description  Add positional or temporal clause

1. ►*at the time*◄ ajax and hercules were trapped behind a landslide at the gaillard cut , both were working to clear the landslide .
2. ►*at the docks ,*◄ hikaru attempts to befriend the tiger , but finds that it dislikes humans .
3. ►*about the second ,*◄ i do know they exist , but the question is whether they are considered a genre outside of japan .
4. ►*in the battle ,*◄ shirou uses his reality marble , unlimited blade works and defeats gilgamesh .
5. ►*in the game ,*◄ red is a curious 11 - year - old boy from pallet town .

Table 10: Example clusters on GithubEdits data using representations learned by a Graph2Tree editor with sequential edit encoder. Locally defined variable names are canonicalized.

| | |
|---|---|
| Description | Switch from `Assert.Equal` to `Assert.Empty` |

```
x-   Assert.Equal(0, V0.ProjectIds.Count);
x+   Assert.Empty(V0.ProjectIds);

x-   Assert.Equal(0, V0.ProjectReferences.Count());
x+   Assert.Empty(V0.ProjectReferences);

x-   Assert.Equal(0, V0.TrustedSelectionPaths.Count);
x+   Assert.Empty(V0.TrustedSelectionPaths);

x-   Assert.Equal(0, V0.Count);
x+   Assert.Empty(V0);

x-   Assert.Equal(0, V0.Messages.Count);
x+   Assert.Empty(V0.Messages);
```

| | |
|---|---|
| Description | Use conditional access |

```
x-   Type V0 = V1 == null ? null : V1.GetType();
x+   Type V0 = V1?.GetType();

x-   V0 = ((V1!= null) ? V1.Operand : null) as MemberExpression;
x+   V0 = V1?.Operand as MemberExpression;

x-   string V0 = V1 == null ? null : V1.GetType().Name;
x+   string V0 = V1?.GetType().Name;

x-   var V0 = V1 == null ? null : V1(V2).ToArray();
x+   var V0 = V1?.Invoke(V2).ToArray();
```

| | |
|---|---|
| Description | Optimize LINQ queries |

```
x-   var V0 = V1.Customers.Where(V2 => V2.CustomerID == LITERAL)
                   .FirstOrDefault();
x+   var V0 = V1.Customers
               .FirstOrDefault(V2 => V2.CustomerID == LITERAL);

x-   var V0 = V1.TypeConverters.Where(V2 => V2.CanConvertTo(V3, V1))
               .FirstOrDefault();
x+   var V0 = V1.TypeConverters
               .FirstOrDefault(V2 => V2.CanConvertTo(V3, V1));

x-   var V0 = this.V1.Where(V2 => V2.CanDeserialize(V3))
                   .FirstOrDefault();
x+   var V0 = this.V1.FirstOrDefault(V2 => V2.CanDeserialize(V3));

x-   var V0 = V1.Where(V2 => V2.Item1 == V3 && V2.Item2 == V4)
                   .FirstOrDefault();
x+   var V0 = V1.FirstOrDefault(V2 => V2.Item1 == V3 && V2.Item2 == V4);
```

| | |
|---|---|
| Description | Change from `Add` function to indexer. |

```
x-   V0.Add(V1.key, V1.V2);
x+   V0[V1.key] = V1.V2;

x-   V0.Add(V1.Id, V2);
x+   V0[V1.Id] = V2;

x-   V0.Add(V1.Etag, V1);
x+   V0[V1.Etag] = V1;

x-   V0.Add(V1.V2, V3.Merge(V1.V4));
x+   V0[V1.V2] = V3.Merge(V1.V4);
```

Table 11: Break-down performance results on the transfer learning task. See Tab. 7 for descriptions of each fixer category.

| Fixer ID | Graph2Tree — Seq Edit Encoder | | | | Seq2Seq — Seq Edit Encoder | | | |
|---|---|---|---|---|---|---|---|---|
| | Acc.(%) | Acc.$^*$(%) | Recall@5(%) | Recall@5$^*$(%) | Acc.(%) | Acc.$^*$(%) | Recall@5(%) | Recall@5$^*$(%) |
| CA2007 | 88.0 | 89.2 | 88.2 | 94.3 | 52.7 | 91.9 | 61.0 | 93.8 |
| IDE0004 | 69.8 | 92.5 | 73.6 | 94.3 | 45.3 | 98.1 | 45.3 | 98.1 |
| RCS1015 | 28.6 | 82.9 | 40.0 | 82.9 | 40.0 | 71.4 | 42.9 | 71.4 |
| RCS1021 | 30.7 | 60.8 | 33.3 | 67.6 | 7.8 | 56.2 | 17.8 | 72.3 |
| RCS1032 | 8.3 | 37.5 | 8.3 | 45.8 | 20.8 | 45.8 | 20.8 | 45.8 |
| RCS1058 | 93.0 | 88.4 | 95.3 | 90.7 | 37.2 | 69.8 | 39.5 | 76.7 |
| RCS1077 | 6.5 | 69.5 | 6.5 | 74.0 | 7.5 | 84.0 | 7.5 | 84.5 |
| RCS1089 | 96.0 | 98.7 | 98.7 | 98.7 | 76.0 | 98.7 | 76.0 | 98.7 |
| RCS1097 | 15.0 | 90.0 | 15.0 | 90.0 | 25.0 | 90.0 | 25.0 | 95.0 |
| RCS1118 | 95.4 | 98.1 | 99.6 | 99.6 | 93.7 | 99.6 | 98.7 | 1.00 |
| RCS1123 | 66.1 | 81.7 | 68.8 | 86.2 | 64.2 | 87.2 | 65.1 | 94.5 |
| RCS1146 | 54.9 | 81.7 | 56.3 | 85.9 | 45.1 | 76.1 | 57.7 | 91.5 |
| RCS1197 | 5.3 | 25.3 | 5.3 | 33.7 | 12.6 | 40.0 | 12.6 | 50.0 |
| RCS1202 | 28.6 | 67.9 | 37.5 | 75.0 | 28.6 | 69.6 | 32.1 | 80.4 |
| RCS1206 | 75.0 | 99.1 | 75.9 | 99.1 | 50.0 | 1.00 | 50.0 | 1.00 |
| RCS1207 | 26.2 | 73.8 | 28.6 | 90.5 | 7.1 | 64.3 | 11.9 | 88.1 |

$^*$: upper-bound performance of predicting $x_+$ using the gold-standard edit representations.

## C  BREAK-DOWN ANALYSIS OF TRANSFER LEARNING RESULTS

Tab. 11 lists the detailed evaluation results for the transfer learning experiments discussed in Sect. 4.4. We refer readers to Tab. 7 for detailed descriptions of each fixer category. The neural Graph2Tree editor outperforms the Seq2Seq editor (both with sequential edit encoders) on 10 out of 16 fixer categories. However, we found that there are categories where both end-to-end system under-performs, even though the upper-bound accuracy is high (*e.g.* RCS1077, RCSRCS1197, RCS1207, RCS1032). While improving the generalization ability of the neural editor models to achieve better transfer learning performance is an important future work, we remark that this task is indeed non-trivial. First, some fixer categories cover a broad range of similar edits, which could not be captured by a single seed edit. xSecond, some categories contain syntactically or semantically complex refactoring rules. For instance, RCS1207 converts method groups into anonymous functions, involving changing multiple positions of the source code, which might not be trivially captured by the sequential edit encoder from a single example edit. Additionally, RCS1197 requires reasoning about a chain of expressions. It turns sb.Append(s1 + s2 + ... + sN) into sb.Append(s1).Append(s2).[...]Append(sN)), which our current models are unable to reason about. More interestingly, we found that there are cases where the edits are syntactically simple, but could be semantically more difficult to learn. For instance, RCS1032 is about removing redundant parentheses from expressions. Although the edit pattern might seem to be syntactically simple at the AST level (replacing a ParethesizedExpressionSyntax node by its child node), determining which pair of parentheses is actually redundant in an expression (*e.g.* (a + b) * (c / d)) is semantically non-trivial to learn from a single edit example. We believe that further advances in (general) learning from source code are required to correctly handle these cases.

## D  IMPACT OF TRAINING SET SIZE

To evaluate the data efficiency of our proposed approach, we tested the end-to-end performance of our neural editor model (Sect. 4.4, Tab. 4) with varying amount of training data. Tab. 12 lists the results. We found both Graph2Tree and Seq2Seq editors are relatively data efficient. They registered around 90% of the accuracies achieved using the full training set with only 60% of the training data.

## E  DETAILS OF HUMAN EVALUATION

As discussed in Sect. 4.2, we performed human evaluation to rate the qualities of neighboring edits given a seed edit. The annotation instructions on GithubEdits and WikiAtomicEdits datasets are listed below. The annotation was carried out by three authors of this paper, and we anonymized the source of systems that generated the output. The three-way Fleiss' kappa inter-rater agreement is $\kappa = 0.55$, which shows moderate agreement (Artstein & Poesio, 2008), an agreement level that is also used in other annotation tasks in NLP (Faruqui & Das, 2018).

Table 12: Test performance of end-to-end experiments with varying amount of training data.

| Training Set Size | Acc.@1 (%) | Recall@5 (%) | PPL per token |
|---|---|---|---|
| GitHubEdits | | | |
| Graph2Tree – Seq Edit Encoder | | | |
| 20% | 43.88 | 50.53 | 1.5703 |
| 40% | 50.44 | 56.63 | 1.4152 |
| 60% | 53.78 | 60.00 | 1.3720 |
| 80% | 55.51 | 60.85 | 1.3392 |
| 100% | **57.49** | **62.94** | **1.3043** |
| WikiAtomicEdits | | | |
| Seq2Seq – Seq Edit Encoder | | | |
| 20% | 42.87 | 48.24 | 1.4123 |
| 40% | 57.72 | 62.31 | 1.1812 |
| 60% | 65.22 | 69.62 | 1.1070 |
| 80% | 68.44 | 73.34 | 1.0751 |
| 100% | **72.94** | **76.53** | **1.0527** |

Table 13: Annotation Instruction for GitHubEdits Data

---

**Rating 2** Semantically and Syntactically Equivalent

---

*The changed constituents in the seed edit and the neighboring edit are applied to the similar positions of the original sentence, serving the same syntactic and semantic role. For example,*

---

**Examples**

- **Seed Edit**
  ```
  x_   var V0 = V1.Where(V2 => V2.Name == LITERAL).Single();
  x_+  var V0 = V1.Single(V2=> V2.Name == LITERAL);
  ```
- **Neighbor**
  ```
  x_   var V0 = V1.GetMembers().Where(V2 => V2.Kind ==
                                          SymbolKind.Property).Single();
  x_+  var V0 = V1.GetMembers().Single(V2 => V2.Kind ==
                                          SymbolKind.Property);
  ```

- **Seed Edit**
  ```
  x_   Type V0 = V1 == null ? typeof(object) : V1.GetType();
  x_+  Type V0 = V1?.GetType() ?? typeof(object);
  ```
- **Neighbor**
  ```
  x_   string V0 = V1 == null ? string.Empty : VAR1.ToString();
  x_+  string V0 = V1?.ToString() ?? string.Empty;
  ```

- **Seed Edit**
  ```
  x_   Assert.True(Directory.Exists(V0) == V1);
  x_+  Assert.Equal(Directory.Exists(V0), V1);
  ```
- **Neighbor**
  ```
  x_   Assert.True(V0.GetString(V0.GetBytes(LITERAL)) ==
                       V1.ContainingAssembly.Identity.CultureName);
  x_+  Assert.Equal(V0.GetString(VAR0.GetBytes(LITERAL)),
                       V1.ContainingAssembly.Identity.CultureName);
  ```

---

**Rating 1** Syntactically or Semantically Related

---

*The seed and neighboring edits share functionally or syntactically similar patterns.*

**Examples**

The following edit is a related edit of the first example above, as it applies the same simplification (`.Where(COND).Func()` to `.Func(COND)`), but for `FirstOrDefault` instead of `Single`:

- **Seed Edit**
  $x_-$  `var V0 = V1.Where(V2 => V2.Name == LITERAL).Single();`
  $x_+$  `var V0 = V1.Single(V2=> V2.Name == LITERAL);`
- **Neighbor**
  $x_-$  `var V0 = V1.Where(V2 => V3.ReportsTo == V2.EmployeeID).FirstOrDefault();`
  $x_+$  `var V0 = V1.FirstOrDefault(V2 => V3.ReportsTo == V2.EmployeeID);`

The following edit is a related edit of the second example above, as it also replaces a ternary expression for null checking with the `?.` and `??` operators:

- **Seed Edit**
  $x_-$  `Type V0 = V1 == null ? typeof(object) : V1.GetType();`
  $x_+$  `Type V0 = V1?.GetType() ?? typeof(object);`
- **Neighbor**
  $x_-$  `var V0 = V1 != null ? V1.ToList() : new List<TextSpan>();`
  $x_+$  `var V0 = V1?.ToList() ?? new List<TextSpan>();`

We also considered pairs such as the following related, since they share similar syntactic structure

- **Seed Edit**
  $x_-$  `V0.State = V1;`
  $x_+$  `V0.SetState(VAR1);`

- **Neighbor**
  $x_-$  `V0.Quantity = V1;`
  $x_+$  `V0.SetQuantity(V1);`

---

**Rating 0** Not Related

---

*The seed and neighboring edits are not related based on the above criteria.*

---

Table 14: Annotation Instruction for WikiAtomEdits Data

| **Rating 2** Semantically and Syntactically Equivalent | |
|---|---|

*The changed constituents in the seed edit and the neighboring edit are applied to the similar positions of the original sentence, serving the same syntactic and semantic role. For example,*

| Seed Edit | Neighbor |
|---|---|
| chaz guest ( born ►*1961*◄) was born in niagra falls , . . . , a decorated hero in wwii in europe , including the purple heart . | randal l. schwartz ( born november 22 , ►*1961*◄) , also known as merlyn , is an american author , system administrator and programming consultant. |
| he was elected to donegal county council for sinn fin in 1979 , and held his seat until his death ►*at age 56*◄ . | davis graduated from high school in january 1947 , immediately enrolling at wittenberg college in rural ohio ►*at age 17*◄ . |
| ►*dror*◄ feiler served as a paratrooper in the israel defense forces . | ►*nagaur*◄ fort - sandy fort ; centrally located ; 2nd century old ; witnessed many battles ; lofty walls & spacious campus ; having many palaces & temples inside . |
| the original old bay house , home of the chief factor , still exists ►*and is now part of the fort vermilion national historic site*◄ . | the population was 6,400 at the 2010 census ►*and is part of the st. louis metropolitan area*◄ . |

| **Rating 1** Syntactically Related | |
|---|---|

*The changed constituents in the seed and the neighboring edit are applied to the similar positions of the original sentence, and they play similar syntactic roles. This includes examples like adding a disfunction, adding a complement, prepositional clause or other syntactic constructs with similar phrases or language structures. For example,*

| Seed Edit | Neighbor |
|---|---|
| the douro fully enters portuguese territory just after the confluence with the gueda river ; once the douro enters portugal , major population centres are less frequent ►*along the river*◄ . | she made a brief return to the screen in " parrish " ( 1961 ) , playing the supporting role of mother which received little attention ►*by the press*◄ . |
| when they found it , they discovered a group of pagumon living there instead who immediately proceeded to treat the digidestined as honored guests ►*, saying that pagumon are the fresh form of koromon*◄ . | in 2012 slote and his baseball book " jake " were the subject of an espn ( 30 for 30 ) short documentary in which slote describes his writing process and reads from the book ►*, saying it is his best writing*◄ . |
| the aircraft was intended to be ►*certified and*◄ supplied as a complete ready - to - fly - aircraft for the flight training and aerial work markets . | in june reinforcements finally did arrive when ►*provincial and*◄ militia units from new york , new jersey , and new hampshire were sent up from fort edward by general daniel webb . |

| **Rating 0** Not Related | |
|---|---|

*The seed and neighboring edits are not related based on the above criteria.*

