# OpenReview forum: "Learning to Represent Edits"
_ICLR.cc/2019/Conference_

### Official Review · AnonReviewer2 · 2018-11-02

**Rating:** 6
**Confidence:** 4

**Review:**

The main contributions of the paper are an edit encoder model similar to (Guu et al. 2017 http://aclweb.org/anthology/Q18-1031), a new dataset of tree-structured source code edits, and thorough and well thought-out analysis of the edit encodings. The paper is clearly written, and provides clear support for each of their main claims.

I think this would be of interest to NLP researchers and others working on sequence- and graph-transduction models, but I think the authors could have gone further to demonstrate the robustness of their edit encodings and their applicability to other tasks. This would also benefit greatly from a more direct comparison to Guu et al. 2017, which presents a very similar "neural editor" model.

Some more specific points:

- I really like the idea of transferring edits from one context to another. The one-shot experiment is well-designed, however it would benefit from also having a lower bound to get a better sense of how good the encodings are.

- If I'm reading it correctly, the edit encoder has access to the full sequences x- and x+, in addition to the alignment symbols. I wonder if this hurts the quality of the representations, since it's possible (albeit not efficient) to memorize the output sequence x+ and decode it directly from the 512-dimensional vector. Have you explored more constrained versions of the edit encoder (such as the bag-of-edits from Guu et al. 2017) or alternate learning objectives to control for this?

- The WikiAtomicEdits corpus has 13.7 million English insertions - why did you subsample this to only 1M? There is also a human-annotated subset of that you might use as evaluation data, similar to the C#Fixers set.

- On the human evaluation: Who were the annotators? The categories "similar edit", and "semantically or syntactically same edit" seem to leave a lot to interpretation; were more specific instructions given? It also might be interesting, if possible, to separately classify syntactically similar and semantically similar edits.

- On the automatic evaluation: accuracy seems brittle for evaluating sequence output. Did you consider reporting BLEU, ROUGE, or another "soft" sequence metric?

- It would be worth citing existing literature on classification of Wikipedia edits, for example Yang et al. 2017 (https://www.cs.cmu.edu/~diyiy/docs/emnlp17.pdf). An interesting experiment would be to correlate your edit encodings with their taxonomy.

---

> ### Author Response · Authors · 2018-11-09
> **Thanks for the comments**
>
> *robustness of edit encodings*: Thanks for the comment! Directly measuring the robustness of edit encodings is non-trivial, but our one-shot learning experiments (Sec. 4.4) serve as a good proxy by testing the editing accuracy using the edit encoding from a similar example.
>
> *applicability to other tasks*: Our proposed method is general and could be applied to other structured transduction tasks. We perform experiment on natural language edits (sequential) and source code commit data (tree-structured), since these are two commonly occurring sources of edits. We leave applying our model to other data sources as interesting future work.
>
> *comparison with Guu et al., 2017*: Thanks for pointing out the related work by Guu et al! As discussed in Section 5, we remark that our motivation and research issues are very different, and these two models are not directly comparable --- Guu et al. focus on learning a generative language model by marginalizing over latent edits, while our work focuses on discriminative learning of (1) representing edits given the original (x-) and edited (x+) data, and (2) applying the learned edit to new input data. We therefore directly evaluate the quality of neighboring edit representations via human annotation, and the end-to-end performance of applying edits to both parallel data and in a novel one-shot learning scenario, which are not covered in Guu et al.
>
> Nevertheless, our model architecture shares a similar spirit with Guu et al. For example, the model in Guu et al. also has an edit encoder based on “Bag-of-Edits” (i.e., the posterior distribution $q(z|x-, x+)$) and a seq2seq generation (reconstruction) model of x+ given x- and the edit representation z. In some sense, our seq2seq editor with a “Bag-of-Edits” edit encoder would be similar as the “discriminative” version of Guu et al. We will make the difference between this research and Guu et al clearer in an updated version of the paper. Please also refer to below for our response to the “Bag-of-Edits” edit encoder.
>
> Response to your specific questions:
>
> *lower-bounding transfer learning results*: Thanks for the comments! Having a lower-bound is helpful in understanding the relative advantage of our proposed method, however it is not clear what a reasonable lower-bounding baseline would be. One baseline would be an editor model (e.g., Graph2Tree with sequential edit encoder) that doesn’t use edit encodings.
>
> *constrained versions of the edit encoder*: First, we remark that our Bag-of-Word edit encoder (Table 1 and 2) is similar to a “Bag-of-Edits” model, where the representation of an edit is modeled by a vector of added/deleted tokens (we use different vocabularies for added and deleted words). Our neural edit encoders have access to the full sequences x- and x+.
>
> We also tried a distributional bag-of-edits model like the one used in Guu et al., using an LSTM to summarize only changed tokens. This model had worse performance in our end-to-end experiment (Table 4) and we therefore we did not include the results. Through error analysis we found that many edits are **context and positional sensitive**, and encoding context (i.e., full sequences) is important. For instance, the WikiAtomicEdits examples we present in Table 9 clearly indicate that semantically similar insertions also share similar editing positions, which cannot be captured by the bag-of-edits encoder as in Guu et al. This might be more obvious for structured data source like code edits (c.f., Table 10). For instance, in the first example in Table 10, `Equal()` can be changed to `Empty()` **only** in the `Assert` namespace (i.e., the context). We apologize for the confusion and will include more results and analysis in the final version, facilitating more direct comparison with the editor encoder in Guu et al. Nevertheless, we remark that as discussed above, our work is not directly comparable with Guu et al.
>
> *subsampling WikiAtomicEdits*: At the time of submission the WikiAtomicEdits dataset could not be downloaded in full, due to an error with the zip file provided. We managed to extract the first 1M edits from the dataset. We believe that the full corpus would not present significantly different statistical properties from the 1M samples we used.
>
> *human evaluation*: please refer to our response regarding annotation. The idea of separating syntactically and semantically similar edits is also very interesting, which we will explore in our final version.
>
> *soft metric*: Thanks for the comment! We can definitely do BLEU evaluation on WikiAtomEdits. For source code data, a sensible “soft” metric on source code still remains an open research issue (Yin and Neubig, 2017). We will include more discussion in our final version.
>
> *classifying Wikipedia edits*: This is a very great idea, thanks for suggesting this. Given the time constraints, we will examine the feasibility of doing something like this for the final version of the paper.

---

> > ### Author Response · Authors · 2018-11-24
> > **Updates to Paper**
> >
> > Thanks again for your insightful comments! We have updated our submission. Below is a brief summary of the changes we made reflecting your comments:
> >
> > **Details of Data Annotation** We included our annotation instructions, and the inter-rater agreement score in Appendix E.
> >
> > **Comparison with the Guu et al. Bag-of-Edits Encoder** As pointed out by you, Guu et al. (2017) introduced a generative language model of natural language sentences by editing prototypes. We have included a more detailed explanation in Section 5 (first Para.) to distinguish our work from Guu et al. (2017). While we remark that our work and Guu et al. (2017) are not directly comparable, we have implemented the deterministic version of the “Bag-of-Edits” edit encoding model in Guu et al. (2017) as a baseline editor encoder for our end-to-end experiments in Section 4.4, Table 4. The results confirm the advantage of our edit encoder models proposed in Section 3.2, which go beyond the simple “Bag-of-Edits” scheme and can capture the context and positional information of edits.
> >
> > As in our previous response to your comment, We have presented further analysis regarding the contextual and positional sensitivity of edits in Appendix B, illustrating the importance of using more advanced edit encoders than "Bag-of-Edits" encoders to capture such information.
> >
> > **“Lower-bounds” of the Transfer Learning Task** We included “lower-bounds” accuracies for the transfer learning experiments in Table 5. To approximate the lower-bounds, we trained Seq2Seq and Graph2Tree transduction models without using edit encoders, and test the model’s accuracies in directly transducing an original input code $x-$ into the edited one $x+$.

---

> > > ### Comment · AnonReviewer2 · 2018-12-07
> > > **Concern about annotation scheme**
> > >
> > > Thank you for the updates!
> > >
> > > In agreement with R3's concerns, I do think it's important to state (prominently) that the annotation was performed by the authors. It seems fairly clear that there are significant qualitative differences, especially between the output of the BoW and seq encoders, and that it would be difficult to avoid bias here. That being said, I think this /does/ reinforce that the differences between models are consistent and measurable.

---

> > > > ### Author Response · Authors · 2018-12-07
> > > > **Human Annotation by Authors**
> > > >
> > > > Thanks for clarifying! We certainly agree that this needs to be stated as prominently as possible and we will make changes to state this more prominently and clearly in the next version of the paper.

---

### Official Review · AnonReviewer3 · 2018-11-04
**This work introduces a new learning task of automated edits for text/code, a learning framework for it, a dataset, and some evaluations but we found mostly the latter lacked, reducing our enthusiasm.**

**Rating:** 6
**Confidence:** 3

**Review:**

The authors state nicely and clearly the main contributions they see in their work (Intro, last paragraph). Specifically the state the paper: 1) present a new and important machine learning task, 2) present a family of models that capture the structure of edits and compute efficient representations, 3) create a new source code edit dataset, 4) perform a set of experiments on the learned edit representations and present promising empirical evidence that the models succeed in capturing the semantics of edits.

We decided to organize this review by commenting on the above-stated contributions one at a time:

“A new and important machine learning task”

Regarding “new task”:

PRO: We are unfamiliar with past work which presents this precise task; the task is new. Section 5 makes a good case for the novelty of this work.

CON: None.


Regarding “important task”:

PRO: The authors motivate the task with tantalizing prospective applications-- automatically editing text and code, e.g. for grammar, clarity, and style. Conceptualizing edits as NLP objects of interest that can be concretely represented, clustered, and used for prediction is an advance.

CON: Many text editors, office suites, and coding IDEs already include features which automatically suggest or apply edits for grammar, clarity, and style. The authors do not describe shortcomings in existing tools that might be better addressed using distributed representations of edits. Consequently, the significance of the proposed contribution is unclear.


“A family of models that capture the structure of edits and compute efficient representations”

Regarding “a family of models”:

PRO: The family of models presented by the authors clearly generalizes: such models may be utilized for computational experiments on datasets and edit types beyond those specifically utilized in this evaluation. The authors apply well-utilized neural network architectures that may be trained and applied to large datasets. The architecture of the neural editor permits evaluation of the degree to which the editor successfully predicts the correct edit given a pre-edit input and a known representation of a similar edit.

CON: The authors do not propose any scheme under which edit representations might be utilized for automatically editing text or code when an edit very similar to the desired edit is not already known and its representation available as input. Hence, we find the authors do not sufficiently motivate the input scheme of their neural editor. The input scheme of the neural editor makes trivial the case in which no edit is needed, as the editor would learn during training that the output x+ should be the same as the input x- when the representation of the “zero edit” is given as input. While the authors discuss the importance of “bottlenecking” the edit encoder so that it does not simply learn to encode the desired output x+, they do not concretely demonstrate that the edit encoder has done otherwise in the final experiments. Related to that: If the authors aimed to actually solve automated edits in text/code then it seems crucial their data contained "negative examples" i.e. segments which require no edits. In such an evaluation one would test also when the algorithm introduces unnecessary/erroneous edits.


Regarding “capture structure of edits”:

PRO: The authors present evidence that edit encoders tightly cluster relatively simple edits which involve adding or removing common tokens. The authors present evidence that relatively simple edits completed automatically by a “fixer” often cluster together, i.e. a known signal is retained in clustering. The authors present evidence that the nearest neighbors of edits in an edit-representation space often are semantically or structurally similar, as judged by human annotators. Section 4.3 includes interesting observations comparing edit patterns better captured by the graph or seq edit encoders.

CON: The details of the human annotation tasks which generated the numerical results in Tables 1 and 2 are unclear: were unbiased third parties utilized? Were the edits stripped of their source-encoder label when evaluated? Objectively, what separates an “unrelated” from a “similar” edit, and what separates a “similar” from a “same” edit? Did multiple human annotators undertake this task in parallel, and what was their overall concordance (e.g. “intercoder reliability”)? Without concrete answers to these questions, the validity and significance of the DCG/NDCG results reported in Tables 1 and 2 are unclear. It is not clear from the two examples given in Table 1 that the three nearest neighbors embedded by the Seq encoder are “better”, i.e. overall more semantically and/or syntactically similar to the example edit, than those embedded by the Bag of Words model. It is unclear which specific aspects of “edit structure” are better captured by the Seq encoder than the Bag of Words model. The overall structure of Tables 1 and 2 is awkward, with concrete numerical results dominated by a spatially large section containing a small number of examples.


“create a new source code edit dataset”

PRO: The authors create a new source code edit dataset, an important contribution to the study of this new task.

CON: Minor: is the provided dataset large enough to do more than simple experiments? See note below on sample size.


“present promising empirical evidence that the models succeed in capturing the semantics of edits”

PRO: The experiment results show how frequently the end-to-end system successfully predicted the correct edit given a pre-edit input and a known representation of a similar edit. Gold standard accuracies of more than 70%, and averaged transfer learning accuracies of more than 30%, suggest that this system shows promise for capturing the semantics of edits.

CON: Due to concerns expressed above about the model design and evaluation of the edit representations, it remains unclear to what degree the models succeed in capturing the semantics of edits. Table 11 shows dramatic variation in success levels across fixer ID in the transfer learning task, yet the authors do not propose ways their end-to-end system might be adjusted to address areas of weak performance. The authors do not discuss the impact of training set size on their evaluation metrics. The authors do not discuss the degree to which their model training task would scale to larger language datasets such as those needed for the motivating applications.

##############
Based on the authors' response, revisions, and disucssions we have updated the review and the score.

---

> ### Author Response · Authors · 2018-11-09
> **Thanks for your comments!**
>
> Thank you for the careful reading of the paper (including the lengthy appendices!), and elucidating concerns about validity of the task and method. We believe that several of these were due to a lack of clarity in our exposition, that can be resolved. We have attempted to clarify these below and will revise the paper to make things more clear before the end of the review period.
>
> * Regarding "important task"
>
> Response: existing editing systems, like the the grammar checker in MS Word and code refactoring module in IDEs, often use heavily engineered, domain-specific, manually crafted rules to perform editing. Our proposed learning-based model is a data-driven approach that automatically **learns** to extract, represent and apply edits from large-scale edit data, and it is also a **generic** system that could be applied to heterogeneous domains like text and source code.
>
> Additionally, using distributed representations also facilitates visualization (Figure 2) and clustering (Appendix B) of semantically similar edits. These novel applications open possibilities to develop human-assistance toolkits for discovering and extracting emerging edit patterns (e.g., new bug fixes from GitHub commits) for rule-based systems from large-scale edit data. For example, this could be used to drive the development of new rules for existing edit tools, by identifying common patterns not covered by existing capabilities. We apologize and will make this clearer.
>
> * Regarding "a family of models"
>
> Response: We agree that our current system is not able to identify places where an edit should be performed, and that this is important future work. In this work, we have focused on (1) computing representations of edits that allow us to group similar changes, and (2) applying such representations in a new context. Both of these scenarios are already useful in human-in-the-loop scenarios. For example, a good solution to problem (1) can inform the development of new edit and refactoring tools (by observing common changes), whereas (2) can be used to propose changes that can be accepted/rejected by a human.
>
> We will make this aspect of future work clearer in the next version of our paper.
>
> * Regarding “capture structure of edits” and Human Evaluation
>
> Response: please refer to our response regarding annotation.
>
> * Regarding “present promising empirical evidence that the models succeed in capturing the semantics of edits” and Results in Table 11
>
> Response: we thank the reviewer for his effort in analyzing the many statistics we present in Table 11! We remark that this task is a transfer learning task is indeed non-trivial. For instance, some fixer categories cover many different types of edits (e.g., RCS1077 (https://github.com/JosefPihrt/Roslynator/blob/master/docs/analyzers/RCS1077.md handles 12 differents ways of optimizing LINQ expressions). In these cases, edits are semantically related (“improving a LINQ expression”), but this relationship only exists at a high level and is not directly reflected to the syntactic transformations required by the fixer.
>
> Other categories contain complex refactoring rules that require reasoning about a chain of expressions (e.g., RCS1197 (https://github.com/JosefPihrt/Roslynator/blob/master/docs/analyzers/RCS1197.md turns sb.Append(s1 + s2 + … + sN) into sb.Append(s1).Append(s2).[...]Append(sN)), which our current models are unable to reason about. We believe that further advances in (general) learning from source code are required to correctly handle theses cases.
>
> We will expand Appendix C with a more fine-grained analysis of the results in Table 11, providing more background on categories whose results deviate substantially from the average.
>
> [Impact of training set size and scalability]: Thanks for the comments! We will discuss this in our final version.

---

> > ### Author Response · Authors · 2018-11-24
> > **Updates to Paper**
> >
> > Thanks again for your insightful comments! We have updated our submission. Below is a brief summary of the changes we made reflecting your comments:
> >
> > 1. (Regarding Data Annotation) We included our annotation instructions, and the inter-rater agreement score in Appendix E.
> >
> > 2. (Regarding "important task" and "a family of models") We added descriptions in Section 2 describing the difference of our proposed neural approach with existing rule-based editing systems and potential downstream applications facilitated by the task. We leave the problem of identifying which edit representation to apply to an input as interesting future work
> >
> > 3. (Regarding “capture structure of edits” and Human Evaluation) We include human evaluation details in Appendix E. We also apologize for the confusion in interpreting Table 1, and have revised its format accordingly. Example 1 in Table 1 shows that the three nearest neighbors returned by the neural editing model are clearly semantically and syntactically relevant to the seed edit (i.e., both the seed edit and the returned neighbors inserted a sentence describing the profession and date of the birth of the topic person), while the nearest neighbors returned by the bag-of-words baseline only rely on surface token overlap, and are not syntactically/semantically similar to the seed edit. We also include discussions about the contextual and positional sensitivity of edits in Appendix B.
> >
> > 4. (Regarding results in Table 11) We expanded Appendix C, presenting more analysis and discussions for some challenging C# fixer categories (RCS1077, RCSRCS1197, RCS1207, RCS1032).
> >
> > 5. (Regarding the Impact of Training Set Size) We evaluated the precision of our neural editor models with varying amount of training data in Appendix D. The results indicate that our proposed approach is relatively data efficient: our Graph2Tree (on GithubEdits) and Seq2Seq (on WikiAtomicEdits) editors achieve around 90% of the accuracies achieved using the full training set with only 60% of the training data.

---

> > > ### Author Response · Authors · 2018-11-26
> > > **Thoughts on updates?**
> > >
> > > Thanks again for your review. We are wondering if our comments have sufficiently addressed your concerns or if there is something that we might have missed.
> > >
> > > Overall, we would kindly ask that you reconsider your rating given the additional experimental results, evaluations and explanation. Alternatively, could you please provide any further guidance on how to improve the paper?

---

### Official Review · AnonReviewer1 · 2018-11-05
**This paper looks at learning to represent edits for text revisions and code changes. The main contribution is in defining a new task, providing a new dataset, and building simple neural network models that show good performance.**

**Rating:** 7
**Confidence:** 3

**Review:**

This paper looks at learning to represent edits for text revisions and code changes. The main contributions are as follows:
* They define a new task of representing and predicting textual and code changes
* They make available a new dataset of code changes (text edit dataset was already available) with labels of the type of change
* They try simple neural network models that show good performance in representing and predicting the changes

The NLP community has recently defined the problem of predicting atomic edits for text data (Faraqui, et al. EMNLP 2018, cited in the paper), and that is the source of their Wikipedia revision dataset. Although it is an interesting problem, it is not immediately clear from the Introduction of this paper what would be enabled by accurate prediction of atomic edits (i.e. simple insertions and deletions), and I hope the next version would elaborate on the motivation and significance for this new task.

The "Fixer" dataset that they created is interesting. Those edits supposedly make the code better, so modeling those edits could lead to "better" code. Having that as labeled data enables a clean and convincing evaluation task of predicting similar edits.

The paper focuses on the novelty of the task and the dataset, so the models are simple variations of the existing bidirectional LSTM and the gated graph neural network. Because much of the input text (or code) does not change, the decoder gets to directly copy parts of the input. For code data, the AST is used instead of flat text of the code. These small changes seem reasonable and work well for this problem.

Evaluation is not easy for this task. For the task of representing the edits, they show visualizations of the clusters of similar edits and conduct a human evaluation to see how similar these edits actually are. This human evaluation is not described in detail, as they do not say how many people rated the similarity, who they were (how they were recruited), how they were instructed, and what the inter-rater agreement was. The edit prediction evaluation is done well, but it is not clear what it means when they say better prediction performance does not necessarily mean it generalizes better. That may be true, but then without another metric for better generalization, one cannot say that better performance means worse generalization.

Despite these minor issues, the paper contributes significantly novel task, dataset, and results. I believe it will lead to interesting future research in representing text and code changes.

---

> ### Author Response · Authors · 2018-11-09
> **Thanks for your comments!**
>
> Question: “what would be enabled by accurate prediction of atomic edits … elaborate on the motivation and significance for this new task”
>
> Response: Our work focuses on developing a generic approach to represent and apply edits. On the WikiAtomicEdits data, one interesting application of our model would be facilitating the development of data exploration toolkits that cluster and visualizes semantically and syntactically similar edits (e.g., the example clusters shown in Table 9). Since our proposed approach is relatively general, we believe we could explore more interesting applications given access to parallel data of other forms of natural language edits. For example, our model could be used to represent and apply syntactic transfer given parallel corpora of sentences with different syntactic structures.
>
> On the source code domain, our work enjoys more intriguing and immediate applications like learning to represent and apply code fixes from commit data, similar to the one-shot learning task we present in Section 4.4. Our work could also enable human-in-loop machine learning applications like clustering commit streams on GitHub at large-scale and helping users identify emerging “best practices” or bug fixes. Indeed, the initial motivation for our research was to automatically identify common improvements to source code that are not covered by existing tools.
>
> Question: "human evaluation is not described in detail..."
>
> Response: please refer to our general response regarding data annotation.
>
> Question: “what it means when they say better prediction performance does not necessarily mean it generalizes better...”
>
> Response: This observation is grounded in the comparison of the results displayed in Tables 4 and 5 in our end-to-end experiment on GitHubEdits data (Section 4.4). Table 4 indicates that given the encoding of an edit (x-, x+), the Seq2Seq editor is most precise in generating x+ from x-, (slightly) outperforming the Graph2Tree editor. We evaluate the generality of edit representations in our “one-shot” experiment, where we use the encoding of a related edit (x-, x+) to reconstruct x+’ from x-’. There, the Graph2Tree editor performs significantly better than the Seq2Seq editor. The latter experiment serves as a good proxy in evaluating the generalization ability of different system configurations, from whose result we derive the hypothesis that better performance with gold-standard edit encodings might not imply better performance with noisy edit encodings.
>
> We apologize for the confusion and will update the text of the paper to clarify what we mean by generalizable and how we draw that conclusion from our experiments.

---

> > ### Author Response · Authors · 2018-11-24
> > **Updates to Paper**
> >
> > Thanks again for your insightful comments! We have updated our submission. Below is a brief summary of the changes we made reflecting your comments:
> >
> > 1. Question: “what would be enabled by accurate prediction of atomic edits … elaborate on the motivation and significance for this new task”
> >      We have presented a detailed explanation in our previous response to your comment. We also revised Section 2 to illustrate some interesting potential downstream applications facilitated by this task. We will include more discussion in the final version given more pages.
> >
> > 2. Question: "human evaluation is not described in detail..."
> >     We included our annotation instructions, and the inter-rater agreement score in Appendix E.
> >
> > 3. Question: “what it means when they say better prediction performance does not necessarily mean it generalizes better...”
> >     We have presented a detailed explanation in our previous response to your comment. We have also rephrased our discussion in Section 4.4 to make the logical flow clearer.

---

### Author Response · Authors · 2018-11-09
**To All Reviewers: Regarding Data Annotation**

To all reviewers:

We thank all reviewers for their insightful comments!

**Regarding Data Annotation**

We apologize for not detailing the annotation rubric and will make this clearer. We will update the main text to clarify the most important points, and provide the instructions and examples for the rating system in the supplementary material.

As also noted by Reviewer-#1, we realized that it is difficult to come up with a fine-grained rating system (e.g., using a 5-element scale) for characterizing semantic/syntactic similarity between edits, especially for free-form natural language data. We believe this problem alone would be an interesting research issue, reminiscent of studies in categorizing syntactic transformations in natural language (e.g., He et al., 2015).

Therefore, we chose to use a simpler 3-element scale (semantically/syntactically equivalent edits, related edits, unrelated). For both natural language and code data, we designed detailed annotation instructions with illustrative examples (will be included in the supplementary material of the next version of our paper). Admittedly, this grading scheme is not perfect, as the category of “relevant edits” could be further divided, and it does not distinguish semantically similar edits from syntactically similar ones. However, we found no way to exactly define how to do such finer-grained annotations, and thus used our simple scheme. Note that this simple grading system is already effective in comparing the performance of different models. For example, we observe a clear win of Seq2Seq models over the bag-of-words baseline in both natural language and code datasets (Tables 1 and 2), and Graph2Tree with sequential edit encoder over Seq2Seq (Table 2), especially in Acc@1.

The annotation was carried out by three of the authors, and we anonymized the source of systems that generated the output. Due to time limits we assigned different sampled edits to different annotators.  We will provide inter-rater agreement score shortly.

Reference:

1. H. He, A. G. II, J. Boyd-Graber, and H. D. III. Syntax-based rewriting for simultaneous machine translation. In Empirical Methods in Natural Language Processing (EMNLP 2015)

---

### Author Response · Authors · 2018-11-24
**TO ALL REVIEWERS: Updates to Paper and Results of Experiments**

We thank again all reviewers for their insightful comments! We have updated our submission reflecting your comments and suggestions. Here is a brief summary of the changes:

**Text Updates**

**Potential Impact of the Task** We revised Section 2, describing the difference of our proposed neural approach with existing rule-based editing systems. We also illustrate some interesting potential downstream applications facilitated by this task.

**Details of Data Annotation** We included our annotation instructions, and the inter-rater agreement score in Appendix E.

**Human Evaluation on WikiAtomicEdits** Based on comments from Reviewer-#3, we revised the format of Table 1 and the corresponding discussion in Section 4.2 to highlight the differences of the neural editing model v.s. a simple bag-of-words baseline.

**Detailed Analysis of the Transfer Learning Experiments** We expanded Appendix C, presenting more analysis and discussions for some challenging C# fixer categories where our model underperformed.


**New Experiments and Analysis**

We also included three new experiments with analysis:

**Comparison with the Guu et al. Bag-of-Edits Encoder** As pointed out by Reviewer-#2, Guu et al. (2017) introduced a generative language model of natural language sentences by editing prototypes. We have included a more detailed explanation in Section 5 (first Para.) to distinguish our work from Guu et al. (2017). While we remark that our work and Guu et al. (2017) are not directly comparable, we have implemented the deterministic version of the “Bag-of-Edits” edit encoding model in Guu et al. (2017) as a baseline editor encoder for our end-to-end experiments in Section 4.4, Table 4. The results confirm the advantage of our edit encoder models proposed in Section 3.2, which go beyond the simple “Bag-of-Edits” scheme and can capture the context and positional information of edits. We also present interesting analysis regarding the contextual and positional sensitivity of edits in Appendix B.

**“Lower-bounds” of the Transfer Learning Task** As suggested by Reviewer-#2, we included “lower-bounds” accuracies for the transfer learning experiments in Table 5. To approximate the lower-bounds, we trained Seq2Seq and Graph2Tree transduction models without using edit encoders, and test the model’s accuracies in directly transducing an original input code $x-$ into the edited one $x+$.

**Performance with Varying Training Data Size** To address the concern raised by Reviewer-#3, we evaluated the precision of our neural editor models with varying amount of training data. We present the results in Appendix D. The results suggest that our proposed approach is relatively data efficient: our Graph2Tree (on GithubEdits) and Seq2Seq (on WikiAtomicEdits) editors achieve around 90% of the accuracies achieved using the full training set with only 60% of the training data.

Finally, we would like to thank again the reviewers for their time and insightful comments which have helped make this paper better. We believe that learning to represent edits is an important yet underexplored problem in representation learning for natural language, source code, and other structured data. We hope that this work inspires future research and that the provided datasets/evaluation protocols will further facilitate future exploration of this task in the community.

---

### Public Comment · (anonymous) · 2019-05-08
**Lack of comparison with state-of-art in SE field**

It seems that the aim of learning representation of source code changes is similar with research work about systematic edits, which is to learn edit script from some example changes and automatically apply the edit script to other code. Therefore, some methods like LASE [1] and Rase [2] used to address systematic edits should be compared.

[1] Meng, N., Kim, M., & McKinley, K. S. (2013). LASE: Locating and Applying Systematic Edits by Learning from Examples. In Proceedings of the 2013 International Conference on Software Engineering (pp. 502–511). Retrieved from http://dl.acm.org/citation.cfm?id=2486788.2486855
[2] Meng, N., Hua, L., Kim, M., & McKinley, K. S. (2015). Does Automated Refactoring Obviate Systematic Editing? In Proceedings of the 37th International Conference on Software Engineering - Volume 1 (pp. 392–402). Retrieved from http://dl.acm.org/citation.cfm?id=2818754.2818804

---

> ### Author Response · Authors · 2019-05-08
> **Differences to learning to apply edits**
>
> Thank you for these pointers. While our paper also describes a component to automatically apply edits, that is only required as part of a system that is able to recognise common edits in a very large set of edits. For example, LASE requires the user to provide edits that can be generalised into a common edit script.
>
> Our paper is concerned with being able to find groups of edits that describe the same change in a very large set of changes (e.g., all of GitHub); and indeed, our core motivation is to use the method from this paper to identify similar edits, and then feed them to a tool that can extract interpretable edit scripts (though we were thinking more of the method of Rolim et al).

---

### Meta-Review · Area_Chair1 · 2018-12-14
**rebuttal improved the review scores, no serious issues other than relatively weak novelty .**

**Confidence:** 2
**Recommendation:** Accept (Poster)

**Metareview:**

This paper investigates learning to represent edit operations for two domains: text and source code. The primary contributions of the paper are in the specific task formulation and the new dataset (for source code edits). The technical novelty is relatively weak.

Pros:
The paper introduces a new dataset for source code edits.

Cons:
Reviewers raised various concerns about human evaluation and many other experimental details, most of which the rebuttal have successfully addressed. As a result, R3 updated their score from 4 to 6.

Verdict:
Possible weak accept. None of the remaining issues after the rebuttal is a serious deal breaker (e.g., task simplification by assuming the knowledge of when and where the edit must be applied, simplifying the real-world application of the automatic edits). However, the overall impact and novelty of the paper is relatively weak.